# A Minireview of the Solid-State Electrolytes for Zinc Batteries

**DOI:** 10.3390/polym15204047

**Published:** 2023-10-10

**Authors:** Wangbing Yao, Zhuoyuan Zheng, Jie Zhou, Dongming Liu, Jinbao Song, Yusong Zhu

**Affiliations:** 1School of Materials Science and Engineering, Anhui University of Technology, Maanshan 243002, China; yiuwangbing@126.com; 2Nanjing Gotion Battery Co., Ltd., Nanjing 211599, China; 3School of Energy Science and Engineering, Nanjing Tech University, Nanjing 211816, China; zhuoyuan@njtech.edu.cn (Z.Z.); jiezhou@ustc.edu.cn (J.Z.)

**Keywords:** rechargeable zinc-ion battery, solid-state electrolyte, Zn-based hybrid-ion batteries, special functional zinc-ion battery

## Abstract

Aqueous zinc-ion batteries (ZIBs) have gained significant recognition as highly promising rechargeable batteries for the future due to their exceptional safety, low operating costs, and environmental advantages. Nevertheless, the widespread utilization of ZIBs for energy storage has been hindered by inherent challenges associated with aqueous electrolytes, including water decomposition reactions, evaporation, and liquid leakage. Fortunately, recent advances in solid-state electrolyte research have demonstrated great potential in resolving these challenges. Moreover, the flexibility and new chemistry of solid-state electrolytes offer further opportunities for their applications in wearable electronic devices and multifunctional settings. Nonetheless, despite the growing popularity of solid-state electrolyte-based-ZIBs in recent years, the development of solid-state electrolytes is still in its early stages. Bridging the substantial gap that exists is crucial before solid-state ZIBs become a practical reality. This review presents the advancements in various types of solid-state electrolytes for ZIBs, including film separators, inorganic additives, and organic polymers. Furthermore, it discusses the performance and impact of solid-state electrolytes. Finally, it outlines future directions for the development of solid-state ZIBs.

## 1. Introduction

As a transmitter of energy from production to consumption, batteries play a role in optimizing time and space allocation in modern society and cover a wide range of applications [1,2]. The most crucial innovator in recent decades has been the rechargeable lithium-ion batteries (LIBs) following their commercialization in 1991 by Sony company, and thereafter, a large range of energy storage applications, such as consumer electronic devices and electronic vehicles, were greatly pumped due to the alleviation of energy anxiety [3]. However, intrinsic issues emerged, including that lithium batteries were plagued by safety concerns as a result of the flammable organic electrolytes [4]. The limited lithium resources have also hindered their further development. 

Aqueous batteries offer several advantages, including low cost, high operating security, and environmental friendliness; they are therefore considered to be one of the most promising alternatives for next-generation batteries [5]. Among various aqueous metal batteries such as Li, Na, K, Mg, Ca, and Al, zinc stands out with remarkable advantages, including low cost, high Zn^2+^/Zn reversibility, and low redox potential (−0.763 V vs. the standard hydrogen electrode) [6,7,8]. Thus, Zn-based aqueous batteries have been rapidly developed in recent years. From the perspectives of cathodes and electrolytes, a wide range of zinc-based aqueous batteries has been proposed, including (1) alkaline zinc batteries [9,10] such as Zn–Ni, alkaline Zn–MnO_2_, Zn–Ag, and Zn–air; (2) neutral or weakly acidic batteries [11], including Zn–MnO_2_, Zn–V_2_O_5_, and electrolytic Zn–Mn batteries; and (3) redox flow batteries [12] such as Zn–Br, Zn–V, Zn–Ce, and Zn–I. However, the anodes for zinc-based batteries are mostly metallic zinc, which possesses some limitations: (1) cathode material dissolution in the electrolytes and the irreversible byproducts of side reactions on electrodes [13]; (2) dendrite formation on zinc anode that causes Coulombic Efficiency (CE) fading and short-circuiting [14]; (3) water splitting reaction that affects output voltage and battery efficiency [15,16]. These serious problems provide direction for research on ZIBs, where the design and development of solid-state electrolytes (SSEs) is a key aspect. 

Among the attempts to overcome the aforementioned drawbacks, SSEs have become an attractive research trend by addressing the problems inherent in liquid electrolytes and rendering advantages due to new battery chemistries [17]. The diverse investigations of SSEs in LIBs inspired the research on Zn-based battery systems. Due to their unique properties, which are different from aqueous systems, solid-state ZIBs have some distinct performances [18]. Compared to the traditional liquid batteries, the internal structures of the solid-state batteries are greatly simplified by replacing the liquid electrolyte, separator, and binder. The SSEs for ZIBs can be generally divided into two groups: the solid-polymer materials and the quasi-solid gel materials. The solid-polymer materials could effectively address the Zn corrosion, Zn passivation, and cathode dissolution issues thanks to their low water content [19]; they also have the benefits of high safety, good mechanical strength, and dendrite inhibition capabilities, whereas their ionic conductivities still need to be improved [20]. In the meantime, quasi-solid gel materials, usually composed of polymer matrices interspersed with salts and solvents (plasticizer) [21], have efficient ion migration channels and reasonable mechanical properties to offer attractive performances [22]: the quasi-solid materials, especially polymer electrolytes, give the battery more mechanical strength, reduced leakage, good flame retardation, high stability in large electrochemical voltage window, and less pressure-related distortion concerns, etc., and are thus used in wearable and human biology applications due to their good flexibility and extended service life [23,24]. Last but not least, the efficiency of their large-scale manufacture can be improved due to their roll-to-roll production manner. However, solid electrolytes need to be improved in the following aspects [25,26]: (1) ionic conductivities at room temperature, (2) electrochemical stabilities, (3) mechanical strength, and (4) interfacial resistance between electrodes and SSEs.

Several review papers on the latest advancements in ZIBs have been published recently; however, these works primarily focus on the development of aqueous-based systems [27,28], the design of electrode active materials [29,30], or the special configurations and applications of ZIBs [31,32]. As of our current understanding, there is no comprehensive overview available that encompasses all types of solid-state zinc secondary batteries. Given the importance of SSEs in advanced energy storage applications, a holistic review of the current advancements in solid-state zinc-ion rechargeable batteries is indeed necessary. Therefore, in this article, first, we systematically summarize the development of solid-state zinc secondary batteries with different electrolyte selections, i.e., ionic liquid, alkaline electrolytes, mildly acidic and neutral electrolytes, and discuss their applications in various electrode systems. Second, the advances in Zn-based hybrid-ion batteries and special functional solid-state ZIBs are analyzed. Finally, the challenges and prospects of solid-state electrolytes used in ZIBs are discussed to explore further research directions.

## 2. Methods for Polymer-Based Electrolyte Preparation

Polymer-based electrolytes play a crucial role in the development of ZIBs, offering advantages such as enhanced safety, improved stability, and increased energy density. The fabrication of polymer-based electrolytes requires advanced preparation methods that enable precise control over electrolyte properties and performance. In this section, we will explore various advanced fabrication approaches and discuss their significance in the preparation of polymer-based electrolytes for ZIBs.

Solution casting is a widely used technique where a polymer is dissolved in a suitable solvent and then cast onto a substrate, forming thin and flexible electrolyte films [33]. This method allows for the formation of electrolytes with controlled thickness and uniformity, with the ability to tailor the physical and electrochemical properties of the electrolyte.

In-situ polymerization involves direct synthesis of the polymer-based electrolyte within the battery system [34]. This technique allows for the fabrication of electrolytes with excellent adhesion to the electrode surfaces and enhanced interfacial stability. In-situ polymerization offers precise control over the polymer structure, composition, and compatibility with other battery components, leading to improved battery performance [35].

Electrospinning is a technique that produces polymer fibers with diameters ranging from nanometers to micrometers [36]. Electrospun polymer-based electrolytes offer a high surface area, porosity, and interconnected pore networks, facilitating ion transport and enhancing electrochemical performance. This method enables the fabrication of flexible and mechanically robust electrolytes with improved ionic conductivity [37].

Cross-linking methods involve the formation of chemical or physical cross-links within the polymer matrix to enhance the electrolyte’s mechanical strength, thermal stability, and dimensional stability [38]. Cross-linking techniques, such as chemical agents, radiation-induced cross-linking, or physical cross-linking through reversible bonds, can be employed to modify polymer-based electrolytes. Cross-linking enhances the electrolyte’s resistance to mechanical deformation and improves its ability to retain ionic conductivity [39].

3D printing, or additive manufacturing, enables the direct creation of three-dimensional objects with precise control over complex geometries [40]. In the context of polymer-based electrolytes, 3D printing allows for the tailoring of electrolyte architecture and properties at a microscale, optimizing ion transport pathways and overall battery performance [41].

Sol-gel methods synthesize materials from a solution or colloidal suspension that undergoes gelation to form a solid network that enables the synthesis of organic–inorganic hybrid electrolyte materials [42,43]. By controlling the composition and processing conditions, researchers can design electrolytes with enhanced mechanical strength, improved thermal stability, and high ionic conductivity.

Freeze casting involves the controlled freezing of a suspension or solution to form a porous structure [44,45]. This technique facilitates the fabrication of porous electrolyte membranes with well-defined pore structures. The resulting freeze-dried electrolyte exhibits high porosity, interconnected pore networks, and tunable pore sizes, enabling efficient ion transport within the electrolyte.

These advanced fabrication approaches offer distinct advantages and can be combined to further enhance electrolyte properties. By selecting the appropriate method and optimizing the fabrication process, researchers can design polymer-based electrolytes with improved conductivity, stability, and safety for efficient and reliable ZIBs. Continued research and development in this field will pave the way for the commercialization of high-performance ZIBs.

## 3. Solid-State Zinc Secondary Batteries with Ionic Liquid and Organic Electrolyte Additives

Room temperature ionic liquids (ILs) are a type of liquid molten salts comprising organic cations and organic/inorganic anions. They offer several advantages compared to other non-aqueous electrolytes, such as low vapor pressure, nonflammability, high thermal stability, wide electrochemical stability window, and relatively high ionic conductivity. ILs have shown capability for reversible deposition and dissolution of zinc, making them a viable option as electrolyte alternatives for rechargeable zinc batteries. Due to their unique properties, IL-based gel polymer electrolytes (GPEs) are considered promising for use in zinc battery systems, replacing conventional electrolytes.

Xu et al. [46] reported a polymer gel electrolyte consisting of a Zn salt, ionic liquid, and PVDF-HFP, demonstrating remarkable mechanical integrity and strength. The polymer gel electrolyte exhibits an impressive ionic conductivity of approximately 10^−3^ S cm^−1^ at room temperature. The ionic liquids (1-ethyl-3-methylimidazolium trifluoromethanesulfante (EMITf) and 1-ethyl-3-methylimidazolium bis(trifluoromethanesulfonyl)imide (EMITFSI)) can dissolve remarkable amounts of zinc salts and are blended with PVDF-HFP polymer to fabricate zinc ion-conducting polymer electrolyte membranes. The Zn salt/ionic liquid/PVDF-HFP polymer gel electrolytes offer several advantageous features, including high ionic conductivity, enabling efficient charge transfer, a wide electrochemical stability window spanning from an anodic stability limit of 2.8 V vs. Zn^2+^/Zn to a cathodic limit at zinc deposition. Additionally, the electrolytes exhibit exceptional thermal stability, maintaining a single-phase behavior in a temperature range as wide as −50–100 °C. They also possess excellent mechanical integrity and strength, contributing to their suitability for practical applications. These electrolytes are also free from any volatile solvents, further enhancing their safety and stability.

Although the IL-enabled GPE possesses satisfactory ionic conductivity at room temperature, the non-uniform deposition of Zinc-ion and the resulting wide growth of dendrites are still problematic [47]. The PVDF-HFP framework incorporating [EMI-TFSI] IL and catalytic copper ions was proposed, where the copper ions were able to accelerate the plating and migration of Zn^2+^, and the IL domain offered high ion mobility [48]. The resulting GPE presented great inhibition of parasitic reactions and dendrite formation, while the ionic conductivity reached 24.32 mS cm^−1^. To search for the most suitable and compatible IL-based electrolyte against various electrode materials for ZIBs, Muhammad et al. screened 50 combinations of ILs using the COSMO-RS simulation tool and compared their dissolution effects on Zn salt [49]. They identified the tetramethyl ammonium cation-based ILs as an appealing candidate, exhibiting low charge transfer resistance and improving the discharge capacity of the battery.

IL can also serve as a useful additive to stabilize the electrode–electrolyte interface and suppress byproducts. Jinbin et al. [50] rationally designed a poly(ionic liquid), poly(1-carboxymethyl-3-vinylimidazolium bromide (PCMVIm), additive to regulate the migration of Zn ions and promote the behaviors of ZIB electrolytes due to its strong interaction with ions, demonstrating the good cycling and rate performances of the batteries. In addition, a novel deep eutectic solvent, consisting of ethylene glycol and zinc trifluorosulfonate (Znotf), was recently developed, achieving both improved plating-stripping reversibility and ion kinetics [51]. Therefore, the ZIB showed a high capacity of 436 mA h g^−1^ and good rate properties. 

Recently developed organic zinc salt electrolytes with large anions exhibited an excellent electrochemical performance by virtue of the reduction in coordinated water surrounding Zn^2+^ and the decrease in solvation. The utilization of zinc trifluoromethanesulfonic acid (Zn(CF_3_SO_3_)_2_) electrolytes in high-performance rechargeable ZIBs has been widely observed. This electrolyte demonstrates excellent cycle performance and high Coulombic efficiency, reaching approximately 100%. 

Niu et al. [52] created a poly(vinyl alcohol)/Zn(CF_3_SO_3_)_2_ hydrogel electrolyte using a simple freeze/thaw technique. This hydrogel electrolyte demonstrated the remarkable capability of self-healing. The fabrication process of the self-healing electrolyte and its self-healing mechanism are shown in Figure 1A. In addition, PVA/Zn(CF_3_SO_3_)_2_ hydrogel exhibits a broader electrochemical window and improved Zn deposition/dissolution kinetics. This hydrogel electrolyte demonstrates a high ionic conductivity of 12.6 S cm^−1^ due to its distinctive 3D porous network structure. Additionally, it possesses excellent self-healing properties, allowing for complete recovery of its electrochemical performance even after multiple cutting and healing cycles. Moreover, the freezing/thawing process enables the integration of all components, including the cathode, separator, and anode, into the hydrogel electrolyte matrix, resulting in all-in-one ZIBs. However, this process still requires filter paper as the separator, probably due to insufficient mechanical strength. Zn(CF_3_SO_3_)_2_ was further incorporated with triazolium-based ionic liquids and lithium bis(trifluoromethanesulfonyl)imide (LiTFSI) to synthesize electrolytes with the high conductivity and electrochemical stable window (up to 6.36 V at 30 °C) [53]. Other than that, various zinc salts with organic anions, e.g., Zn(TFSI)_2_ [54], zinc trifluoroacetate (Zn(TFA)_2_) [55], zinc acetate (Zn(CH_3_COO)_2_) [56], and Zn(OTf)_2_ [57] have been proposed to exhibit promising performances.

## 4. Solid-State Zinc Secondary Batteries with Alkaline Electrolytes

Alkaline electrolytes have a long history of application in zinc-ion batteries [60]. Alkaline electrolytes offer several advantages compared to neutral and acidic electrolytes. The benefits of alkaline electrolytes encompass a high operating voltage, rapid reaction kinetics, and enhanced ionic conductivity. KOH is the preferred alkaline electrolyte in rechargeable zinc batteries due to its notable characteristics such as the high solubility of zinc salt in KOH solution and the superior ionic conductivity of K^+^ (73.5 S cm^−2^) compared to Na^+^ (50.1 S cm^−2^) and Li^+^ (38.7 S cm^−2^). However, it is important to note that by-products such as ZnO and Zn(OH)_2_ can be generated due to over-discharging. High-concentration solutions can increase the solubility of by-products and improve the electrochemical reaction kinetics. To prevent a decline in the ionic conductivity of the electrolyte, a KOH solution with a concentration of 6 M is commonly utilized. However, zinc dendritic formation, cathode material dissolution, and corrosion of the current collector are more likely to occur in the highly concentrated alkaline electrolytes [14], which will greatly increase the potential safety hazards of the ZIBs in the event of leakage; therefore, solid-state electrolytes may have positive effects. The subsequent discussion focuses on the recent advancements in alkaline electrolyte-based ZIBs featuring various electrode materials.

### 4.1. Zn–MnO_2_ Batteries

Zhang et al. [61] investigated the PVA-based gel electrolyte containing different wt.% KOH to Zn/MnO_2_ batteries. The conductive characteristics of the gel electrolytes closely resemble those of KOH aqueous solutions. Initially, the conductivity increased with an elevation in KOH concentration due to the reduction in the crystalline phase but then decreased at high KOH concentrations because of the restricted ionic mobility [62]. The resulting battery achieved good cycling performance.

Zhu et al. [63] fabricated an elastic PGE film using a solution polymerization method consisting of 0.02 wt.% K_2_S_2_O_8_, 16.75 wt.% acrylic acid, and 83.23 wt.% aqueous KOH solution, which was optimized for the reaction. The utilization of PGE electrolytes in Zn/Air, Zn/MnO_2_, and Ni/Cd cells showed that the PGE film had chemical and electrochemical stability comparable to that of aqueous alkaline solutions.

In a study conducted by Gaikwad et al. [64], a stretchable MnO_2_-zinc cell was fabricated using a gel electrolyte based on polyacrylic acid (PAA). The cell utilized off-the-shelf compliant silver fabric as a current collector, which was embedded with MnO_2_ and Zn particles. Remarkably, the cell demonstrated a discharge capacity of 3.775 mAh cm^−2^ that was fully maintained even under a high strain of 100%.

### 4.2. Zn–Ni Batteries

Lee et al. [65] proposed a poly(acrylamide-co-acrylic acid) gel electrolyte for the Zn–Ni secondary battery. The gel electrolyte was prepared through a straightforward process by dissolving P(AAm-co-AAc) in an alkaline solution and subsequently gelling the mixture. Considering the conductivity and viscosity characteristics of the gel polymer electrolyte, it was observed that the ionic conductivity slightly decreases with an increase in P(AAm-co-AAc) concentration, while the viscosity increases with an increase in P(AAm-co-AAc) concentration. To optimize the gel electrolyte, the concentration of P(AAm-co-AAc) was fixed at 6% by weight, resulting in a conductivity of 4.8 × 10^−1^ S cm^−1^. The utilization of the gel polymer electrolyte significantly improved the capacity retention of the cell. After 60 cycles, the capacity retention reached about 88% (310 mAh g^−1^) in comparison to approximately 40% retention (125 mAh g^−1^) observed with a regular alkaline electrolyte Ni-Zn cell under the same cycling conditions. This highlights the superior performance of the gel polymer electrolyte in enhancing the stability and capacity retention of the Zn–Ni secondary battery. The improved cycling performance was attributed to the lower reactivity of polymer electrolytes compared with liquid electrolytes and the suppression of dendrite growth by gel polymer electrolyte as well as the zinc dissolution confinement in alkaline environments by gel polymer electrolytes.

Xinying et al. [66] developed a 3D cross-linked GPE consisting of poly(acrylamide-potassium acrylate) (P(AM-KA)), zinc alginate, and KOH. Benefiting from the covalent-bonded network and the abundant ion ligand complex, the GPE has favorable ion transport and adsorption, effectively mitigating dendrite growth in the Ni-Zn battery. As a result, the battery showed remarkable stability over 10,000 cycles with a capacity retention of 88.96%.

### 4.3. Zn–Air Batteries

Metal–air batteries have attracted great interest as promising energy storage technologies with distinct energy density advantages, among which the Zn–air technology has been mainly focused on due to its low operation cost, safety, and better striping/plating ability. The most commonly used configuration for a zinc–air battery consists of a zinc anode, an alkaline electrolyte, and an air cathode, typically made from a porous and carbonaceous material. Concentrated aqueous alkaline solutions have been used as the electrolyte by virtue of their better kinetics and catalytic activity, especially KOH, which shows better properties of high ionic conductivity, high activity, large oxygen diffusion coefficient, good low-temperature performance, and good solubility of carbonate by-products [67,68,69]. In the process of discharging a zinc–air battery, oxygen molecules permeate the porous air cathode and undergo reduction to hydroxyl ions, facilitated by the catalyst layer on the cathode. At the same time, electrons are generated through the electrochemical oxidation of zinc. During the recharge process, oxygen is evolved and diffuses out of the cathode, while Zn^2+^ are deposited back onto the anode. In rechargeable batteries, it is crucial to have an oxygen electrode that exhibits dual catalytic activity for both oxygen reduction reaction (ORR) and oxygen evolution reaction (OER), whereas primary zinc–air batteries, which are not designed for recharging, typically employ ORR electrocatalysts that are specialized for one specific reaction. Solid-state zinc–air batteries exhibit better cycling performance since the mitigation of aqueous electrolyte volatilization improves access to oxygen on cathodes, which increases the transportation rate in the cathode. Moreover, polymer electrolytes offer a wider electrochemical stability window and enhanced mechanical robustness. These properties contribute to an extended battery shelf life and an expanded operating temperature range. Therefore, besides the common required properties like high ionic conductivity, the electrolyte should also cut down the content of volatile solvents. Vassal et al. [70] were the pioneers in introducing a potassium hydroxide-based alkaline solid polymer electrolyte, specifically a copolymer of poly(epichlorohydrin) and poly(ethylene oxide) known as P(ECH-co-EO), into zinc–air cells. This innovative electrolyte enabled the cell to deliver a high current density of 14 mA cm^−2^ at a discharge voltage of 0.8 V. Additionally, it demonstrated exceptional performance by sustaining a current density of 30 mA cm^−2^ at the same voltage and operating at a temperature of 60 °C, surpassing the limitations of PEO/KOH/H_2_O electrolytes that tend to melt at this temperature.

PVA is one of the most commonly used polymer electrolytes in solid-state zinc–air batteries thanks to its excellent chemical stability, hydrophilicity, high dielectric constant, and high alkali tolerance [71]. The high humidity absorption of PVA due to –OH groups promotes salt solvation and thereby enhances ionic conductivity [72,73,74]. 

Liu et al. [58] investigated the development of flexible al-solid-state Zn–air batteries using various components such as a free-standing nano-porous carbon nanofiber film-based air cathode, zinc foil anode, alkaline PVA gel electrolyte, and pressed nickel foam current collector (to enhance conductivity). Although the PVA gel electrolyte had limited ionic conductivity and caused high contact resistance, negatively impacting the charge–discharge performance, the battery demonstrated excellent flexibility and cycling stability. The results shown in Figure 1B indicate that the all-solid-state battery displayed consistent charge (1.78 V) and discharge (1.0 V) potentials throughout a 6-h period while operating at a current density of 2 mA cm^−2^. Notably, the battery maintained this stability even when subjected to substantial bending or folding, demonstrating its robustness and flexibility. Yue et al. [75] synthesized a double network hydrogel using agar, graphene oxide, and PVA for Zinc–air batteries to demonstrate both good mechanical strength (388 kPa) and ionic conductivity (75 mS cm^−1^). Furthermore, an agarose biopolymer matrix was developed to directly dissolve in KOH solution, thus avoiding the use of petroleum-based plastics [76].

However, PVA-based electrolytes are often plagued by very poor mechanical properties and insufficient ion-transport capability, which harms the electrochemical performance and mechanical flexibility. Zhi’s group [77] designed an alkaline-tolerant dual-network PANa-cellulose hydrogel through simple radical polymerization. The battery exhibited remarkable properties of super-stretchability (stretched up to 800% in a flat shape and 500% in a fiber-shaped configuration) and a high-power density of 108.6 mW cm^−2^, which was superior to batteries with PVA electrolytes because of the better ionic conductivity when soaked in 6 M KOH. Joohyuk Park et al. [78] used gelatin as an electrolyte for zinc–air cells, and the electrolyte had comparable and higher ionic conductivity even at lower KOH concentrations (0.56 wt.%) compared with previously reported KOH-based GPEs. The gelatin electrolyte met the high requirement of robustness in cable-type flexible zinc–air batteries. Similarly, a cross-linked double-network GPE with carboxymethyl chitosan, acrylamide, and sodium acrylate was proposed, showing a good electrochemical property over a wide temperature range (−20–80 °C) [39].

Other studies on solid-state alkaline zinc rechargeable batteries are summarized in Table 1.

## 5. Solid-State Zinc Secondary Batteries with Mild Acidic and Neutral Electrolytes

Shoji et al. [87] were the first to report the development of a rechargeable aqueous Zn–MnO_2_ battery, in 1988, utilizing a mild neutral or slightly acidic electrolyte. This electrolyte was specifically ZnSO_4_ in an aqueous solution. Since then, rechargeable batteries have experienced rapid development. The utilization of neutral or mildly acidic aqueous electrolytes in zinc-based batteries offers several appealing benefits. These electrolytes help prevent the formation of undesired by-products, minimizing the formation of zinc dendrites, reducing the corrosion of zinc anodes, and enhancing overall safety characteristics. Mild acidic and neutral electrolytes can be more compatible with polymer materials, providing higher possibilities for solid-state electrolytes.

### 5.1. PVA-Based Electrolyte

#### 5.1.1. MnO_2_ Cathode

Despite their high capacity and energy density, Mn-based materials face limitations in their electrochemical performance due to inherent low electronic conductivity and inevitable manganese dissolution. These factors directly result in poor rate capability and rapid capacity fading, thereby posing challenges to their practical applications. The cycle performance of MnO_2_ as a cathode material in aqueous zinc batteries has been subject to criticism. However, a notable improvement can be achieved by introducing MnSO_4_ into the electrolyte. This addition effectively reduces the dissolution of the cathode material and enhances the overall life cycle of the battery [88].

The development of a quasi-solid-state Zn–MnO_2_@PEDOT battery with a gel electrolyte composed of PVA/LiCl–ZnCl_2–_MnSO_4_ was reported by Lu et al. [89]. In general, batteries that utilize quasi-solid-state electrolytes tend to exhibit poorer rate capability compared to those using aqueous electrolytes. This can be attributed to the higher charge transfer resistance associated with polymer electrolytes. Despite the inherent limitations of quasi-solid-state electrolytes, the Zn–MnO_2_@PEDOT battery demonstrated remarkable rechargeability. After 300 cycles, it retained over 77.7% of its initial capacity and achieved nearly 100% Coulombic efficiency. The improved cycling stability can be attributed to the presence of the PEDOT buffer layer and the Mn^2+^-based neutral electrolyte, which effectively suppressed structural pulverization and dissolution of MnO_2_. Furthermore, this quasi-solid-state battery exhibited impressive mechanical flexibility and performed excellently across a wide range of ambient temperatures.

Li et al. [90] employed a PVA/ZnCl_2_/MnSO_4_ gel electrolyte to fabricate a quasi-solid-state Zn–MnO_2_ battery. The construction involved utilizing MnO_2_ nanorod arrays as the free-standing cathode and uniformly depositing tiny Zn nanoparticles on N-doped porous carbon cloth as the anode. The quasi-solid-state battery maintained good cycling performance compared to aqueous electrolytes and achieved a high energy density of 440 Wh kg^−1^ and a high power density of 7.9 kW kg^−1^. While the flexibility and durability of energy storage systems are crucial for advancing the development of flexible and wearable devices, it is important to note that excessive deformation can still cause damage, resulting in performance degradation and potential safety hazards [91]. The mechanical and safety properties of the battery can be improved by taking advantage of the self-repairing function present in the numerous hydroxyl groups and the hydrogen bonds within the PVA segments [92].

#### 5.1.2. Prussian-Blue-Analog-Based Cathode

Prussian blue analogs (PBAs) possess a basic cubic structure in which iron(III) ions are surrounded octahedrally by nitrogen atoms, while iron(II) ions are surrounded by carbon atoms and can be further roughly denoted as A_x_M_Ay_[M_B_(CN)_6_]_z_·nH_2_O, where A is alkali metal and M_A_ and M_B_ are transition metals such as Mn, Fe Co, Ni, Cu, and Zn. The 3D porous framework structures of PBAs offer a multitude of reaction sites and diverse valence states and contribute to their structural stability. The 3D zinc hexacyanoferrate (ZnHCF) nanocubes were proposed by Lu et al. utilizing an in situ co-precipitation method [93] and were then encapsulated with 2D manganese oxide nanosheets (ZnHCF@MnO_2_) and ZnSO_4_/PVA gel electrolyte to assemble a flexible quasi-solid ZIB. The distinct structure of the cathode material includes the implementation of a multi-layer pseudocapacitive MnO_2_ as a buffer layer. This design approach effectively mitigated diffusion-controlled limitations, allowing for efficient regulation of the charge storage process. As a result, it facilitated high operating voltages of approximately 1.7 V and exhibited excellent electrochemical performance, with a capacity of 118 mAh g^−1^ at 100 mA g^−1^ and 75 mAh g^−1^ at 1000 mA g^−1^. 

### 5.2. PAM-Based Electrolytes

#### 5.2.1. MnO_2_ Cathode

Zhi et al. [59] put forward a cross-linked polyacrylamide (PAM) hydrogel electrolyte and constructed a compressible Zn–MnO_2_ battery. This battery demonstrated excellent durability properties, including favorable charge–discharge characteristics, specific capacity, and open-circuit voltage (OCV) even when subjected to compressive loads, as depicted in Figure 1C. The energy storage capacity of the battery improved as the strain increased, which can be attributed to the enhanced contact between the electrode and electrolyte, along with the improved ionic conductivity of the PAM hydrogel. In addition, Zhi’s group utilized the PAM hydrogel electrolyte to create a stretchable yarn ZIB by employing double-helix yarn electrodes. This design resulted in a high specific capacity and volumetric energy density of 302.1 mAh g^−1^ and 53.8 mWh cm^−3^, respectively. Additionally, the battery exhibited excellent cycling stability, with 98.5% capacity retention after 500 cycles [94]. 

Despite the high ionic conductivity exhibited by the PAM hydrogel, its mechanical properties significantly deteriorate when the water content reaches 90%. To address this issue, Zhi et al. [95] proposed the incorporation of a nanofibrillated cellulose (NFC) additive to enhance both the mechanical properties and ionic conductivity of PAM-based electrolytes. By utilizing a free radical polymerization method, PAM forms within the cellulose network and NFC itself easily forms a spacious 3D network [96]. In this polymer, the gel NFC acts as a support for the pore walls, while PAM expands the pore size due to its high water absorbency [97]. As a result, the strength is improved to 158 kPa, four times higher than that of pure PAM, and can withstand strains of up to 1400%. This enhancement can be attributed to the confinement offered by the cellulose nanofiber network. Cellulose also stabilized the porous structure and enlarged the pore size, increasing the ionic conductivity to 22.8 mS cm^−1^ (while pure PAM electrolytes showed an ionic conductivity of 16.9 mS cm^−1^). The sewable solid Zn–MnO_2_ battery obtained in the study retained 88.5% of its initial capacity even after 120 stitches and demonstrated resistance to high shear forces exceeding 43 N. Additionally, when tested at a rate of 4 C, the battery showed a commendable capacity retention of 88.3% after 1000 cycles, along with a high specific capacity of around 200 mAh g^−1^.

Zhi’s team implemented a free radical polymerization process of acrylamide monomers to incorporate PAM onto gelatin chains. They then proceeded to gel the resulting composition on an electrospun PAN membrane, as shown in Figure 2A [98]. An α-MnO_2_ nanorod/carbon nanotube (CNT) material was used as a cathode. The novel hierarchical polymer electrolyte (HPE) film based on gelatin and PAM showed an incredibly high ionic conductivity of 1.76 × 10^−2^ S cm^−1^ at room temperature. The presence of abundant hydrophilic groups facilitated a high water content and the open channels within the 3D hierarchical architecture allowed for uninterrupted ionic pathways in the hydrogel network, thus contributing to the exceptional ionic conductivity. Moreover, the hierarchical polymer film displayed a strength of 7.76 MPa, approximately six times greater than that of the gelatin electrolyte. The solid-state ZIB exhibited an impressive areal energy density of 6.18 mWh cm^−2^ and a commendable specific capacity of 306 mAh g^−1^ at a current density of 61.6 mA g^−1^. Moreover, the ZIB displayed excellent capacity retention, maintaining 97% of its capacity even after 1000 cycles at a higher current density of 2772 mA ^−1^.

A unique design of a molecular network based on P(PEGMEA-AM)/PAM was invented to target the Zn salt dissolution issue [99]. The numerous functional groups on the polymer chains could interact with Zn^2+^, enabling a high ionic conductivity of 82.65 mS/cm and transference number of 0.82 and preventing interfacial polarization. Therefore, the Zn//MnO_2_ cell exhibited a CE of 98.9% and a capacity of 300 mAh/g.

**Figure 2 polymers-15-04047-f002:**
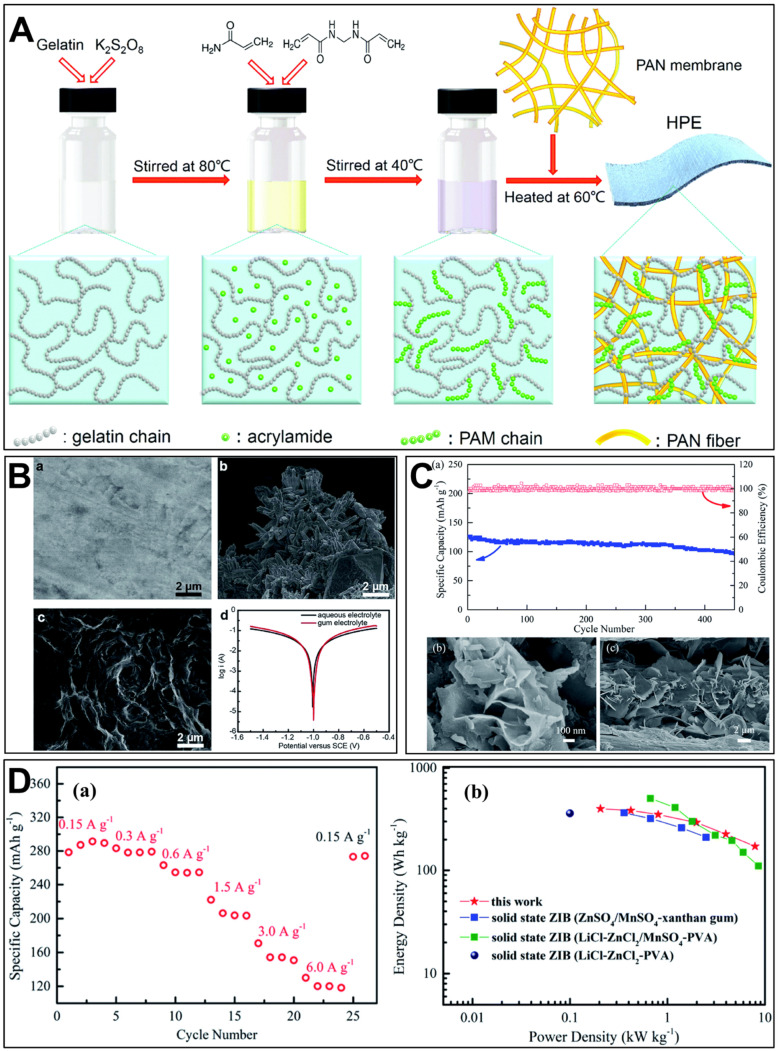
(**A**). Schematic of the synthesis process of the innovative hierarchical polymer electrolyte (HPE) film based on gelatin and PAM. The HPE was created by grafting PAM onto gelatin chains that were then embedded within the network of a PAN electrospun fiber membrane using a straightforward free radical polymerization method. Reproduced from Ref. [98] with permission from the Royal Society of Chemistry. (**B**). SEM images of (**a**) the fresh Zn foil and (**b**) the Zn foil in the battery after 50 cycles at 5 C; SEM images of (**c**) the Zn foil in the xanthan gum electrolyte after 1000 cycles at 5 C; (**d**) Tafel plots of the Zn anode in aqueous and gum electrolytes. Reproduced from Ref. [100] with permission from the Royal Society of Chemistry. (**C**). (**a**) Cycling stability of the solid-state ZIBs with KCR electrolyte cycled at 6.0 A g^−1^, along with the Coulombic efficiency. SEM images of (**b**) the MnO_2_ cathode and (**c**) the electroplated Zn anode after 450 charge–discharge cycles. Reproduced from Ref. [101] with permission from the Royal Society of Chemistry. (**D**). (**a**) Specific capacities of the solid-state ZIBs at various current densities, and (**b**) Ragone plots of the solid-state ZIBs. Reproduced from Ref. [101] with permission from the Royal Society of Chemistry.

#### 5.2.2. Vanadium-Based Cathode

The cost-effectiveness, abundant availability, and multiple oxidation states of vanadium make vanadium oxide and its derivatives highly desirable for various battery system applications. Nazar et al. [102] synthesized a cathode material called layered vanadium oxide bronze (Zn_0.25_V_2_O_5_·*n*H_2_O). The structure of this material was stabilized by interlayer metal ions (Zn^2+^) and structural water molecules. This cathode material exhibited a remarkable capacity of up to 300 mAh g^−1^. Yan et al. [103] conducted a comprehensive study on the influence of structural water on the storage performance of Zn^2+^ in hydrated V_2_O_5_·*n*H_2_O. A “lubricant” effect was proposed, where water molecules accelerated the transport of Zn^2+^ and the charge shielding effect of water crystal molecules could reduce the effective charge of the inserted Zn^2+^, thereby improving the electrochemical performance. Various vanadate-based cathode materials have been developed, including Zn_3_V_2_O_7_(OH)_2_·2H_2_O, Ca_0.25_V_2_O_5_·*n*H_2_O, LiV_3_O_8_, NaV_3_O_8_·1.5H_2_O, Na_2_V_6_O_16_·1.63H_2_O, Na_1.1_V_3_O_7.9_, Na_0.33_V_2_O_5_, Na_5_V_12_O_32_, K_2_V_8_O_21_,Zn_2_V_2_O_7_, Mo_2.5+y_VO_9+z_, Li*_x_*V_2_O_5_·*n*H_2_O, Ag_0.4_V_2_O_5_, V_2_O_5_, VO_2_, and V_3_O_7_·H_2_O (H_2_V_3_O_8_) [13,85,104,105]. The majority of these materials demonstrate notable characteristics such as high specific capacity, excellent rate capability, elevated energy density, and prolonged life cycle.

A recent study conducted by Deng et al. [106] addressed the issue of zinc dendrite formation by utilizing high-capacity layered Mg_0.1_V_2_O_5_·H_2_O (MgVO) nanobelts in combination with a concentrated 3 M Zn(CF_3_SO_3_)_2_ PAM gel electrolyte. By implementing this approach, Deng et al. successfully mitigated the formation of zinc dendrites and achieved a durable and practical ZIB system. This system exhibited ultrahigh capacity, and high-rate capability, and maintained a wide operating temperature range, all contributing to the overall success of the approach. Zhuoyuan et al. [33] created a composite PE based on agar and PAM with a non-porous morphology and high mechanical stability. The corresponding Zn//V_2_O_5_ cell presented not only good cyclic behavior but also excellent adaptability to the changes in ambient temperature. Additionally, a PAM/cellulose PE was innovated to protect the Zn metal electrode due to the good water-retaining phenomenon [107].

### 5.3. Polysaccharide-Based Electrolytes

Zhang et al. [100] mixed xanthan gum with aqueous sulfate solutions to fabricate a sulfate-tolerant gum bio-electrolyte. Different from some commonly used polymer solutions such as poly(ethylene oxide) (PEO), PVA, agar gelatin, and sodium polyacrylate, where adding ZnSO_4_/MnSO_4_ salts can lead to precipitation, xanthan gum solution formed a stable and homogenous electrolyte, indicating its good salt tolerance. The xanthan gum achieved high ionic conductivity (1.46 × 10^−2^ S cm^−1^). The xanthan gum electrolyte was observed to effectively inhibit the growth of Zn dendrites, particularly at heterogeneities present on the surface of the electrode. Furthermore, the corrosion current of Zn foil in the xanthan gum electrolyte demonstrated an 80% reduction compared to that observed in the aqueous electrolyte (2.02 mA) (Figure 2B), which indicated the probable corrosion inhibition effect of xanthan gum on Zn foils, helping to maintain the smooth surface of zinc anode. The implementation of this solid electrolyte in assembled flexible Zn–MnO_2_ batteries resulted in remarkable characteristics, including high specific capacities (260 mAh g^−1^ at 1 C), excellent rate performance, outstanding cycling performance (127 mAh g^−1^ over 1000 cycles at 5 C), and exceptional mechanical robustness. Similarly, a polysaccharide-based hydrogel based on acrylamide and xanthan gum was proposed to form a dual network that restricts the movement of water molecules and promotes ionic conductivity, even at low temperatures [108].

Chunyi Zhi’s group combined kappa-carrageenan with rice paper scaffold to build an electrolyte used for quasi-solid state ZIBs [101]. At room temperature, the KCR electrolyte demonstrated exceptional flexibility and displayed impressive ionic conductivity of 3.32 × 10^−2^ S cm^−1^. The morphological analysis of the cathode and anode in solid-state ZIBs after 450 cycles is illustrated in Figure 2C. The nanosheet structure of MnO_2_ remained unaltered, while the presence of freestanding Zn nanosheets was observed after 450 cycles. This observation suggests that the morphology of the cathode and anode materials in biopolymer cells was effectively maintained. These findings indicate that the morphology of the cathode and anode materials in biopolymer cells was successfully preserved. This suggests that kappa-carrageenan electrolytes improved the stability of battery electrodes and the solid-state battery. Based on the advantages mentioned above, the utilization of a MnO_2_/rGO composite cathode and zinc metal anode in a quasi-solid ZIB resulted in remarkable energy density and power density values of 400 Wh kg^−1^ and 7.9 kW kg^−1^, respectively. Additionally, ZIB exhibited a high specific capacity of 291.5 mA h g^−1^ at 0.15 A g^−1^, along with rapid charging and discharging capabilities of 120.0 mA h g^−1^ at 6.0 A g^−1^, as shown in Figure 2D. 

To obtain the guar gum electrolyte with excellent flexibility and mechanical robustness, Hang Zhou’s group dissolved guar gum in a solution of 2 M ZnSO_4_ and 0.1 M MnSO_4_, which was then solidified [109]. The concentration of 2 M ZnSO_4_ + 0.1 M MnSO_4_ resulted in the highest ionic conductivity at room temperature, reaching a value of 1.07 × 10^−2^ S cm^−1^. By fitting the Arrhenius equation linearly, the lowest activation energy (Ea = 0.0541 eV) was determined. The battery utilizing guar gum demonstrated excellent rate performance, with a capacity of 308.2 mAh g^−1^ at 0.3 A g^−1^ and 131.6 mAh g^−1^ at 6.0 A g^−1^. This remarkable performance can be attributed to the high ionic conductivity of the electrolyte and the conductivity enhancement provided by rGO in the cathode. In addition, the guar gum electrolyte effectively prevented the formation of zinc dendrites during cycling, resulting in an outstanding cycling performance. The battery exhibited 100% capacity retention after 1900 cycles and 85% capacity retention after 2000 cycles at 6.0 A g^−1^.

### 5.4. Alginate-Based Electrolyte

#### 5.4.1. MnO_2_ Cathode

Chunyi Zhi et al. [110] used Zn-alginate as a second ionic network to be introduced in a covalently crosslinked polyacrylamide (PAAm) framework. Then, the as-acquired dual-crosslinked hydrogel electrolyte was used to assemble a mechanically durable Zn–MnO_2_ battery, which demonstrated outstanding mechanical stability and durability due to the effective energy dissipation of the hydrogel under dynamical deformation. Jiang Zhou et al. [111] changed sodium alginate (SA) into zinc alginate gel via a direct ion crosslinking method. The zinc alginate gel exhibited unique ion transfer capability by providing a “pass-way” for Zn^2+^. The migration of ions was guided and controlled by the presence of carboxylate groups, which restricted their movement within the gel [112]. The ion-confinement effect plays a crucial role in suppressing dendrite growth, achieving uniform zinc deposition, and inhibiting side reactions. Furthermore, it demonstrated a remarkable conductivity of 1.83 × 10^−2^ S cm^−1^ at room temperature. The Zn/Alg-Zn/MnO_2_ solid-state batteries exhibited an impressive specific capacity of over 300 mAh g^−1^ at a rate of 0.2 A g^−1^. Additionally, these batteries demonstrated reasonable rate capability. Despite the common issues associated with long-term storage of ZIBs, such as parasitic reactions between electrolyte and electrodes and ion self-diffusion, impressive capacity restoration was observed. After a 60-h resting period, the capacity rapidly recovered and remained at approximately 320 mAh g^−1^ [113]. Zinc alginate gel effectively resulted in a smooth morphology on the surface of the Zn anode and restricted ion transportation, slowing down the parasitic reaction and active Zn consumption, thus delivering a satisfactory shelf-life and restoration capacity. 

In order to address dendritic growth issues on zinc anodes and enhance the cyclic performance of Zn–MnO_2_ batteries, Xueyuan Li et al. [114] synthesized a quasi-gel composite electrolyte by introducing fumed silica (SiO_2_) and SA. The research group proposed the underlying mechanisms as: the three-dimensional network structure composed of SA and SiO_2_ had a function similar to the porous coating or porous fiber layer [115,116], which could guide the uniform migration of Zn^2+^, avoiding the concentration of Zn^2+^ polarization near the dendrite tip during the process of zinc deposition. 

#### 5.4.2. Vanadium-Based Cathode

Kevin Huang et al. [117] introduced a novel three-dimensional (3D) hydrogel electrolyte that was double-crosslinked and derived from gelatin (GE) and SA. This innovative approach combined the hydrophilic properties of GE with the high dielectric strength and modulus of SA. The 3D network structure of the hydrogel electrolyte contained a wealth of polar groups, such as carboxyl and hydroxyl groups, which facilitated the dissociation of inorganic salts and immobilized the electrolyte solution. Simultaneously, the dual network exhibited strong covalent bonding as well as reversible ionic and hydrogen bonding. These unique characteristics endowed the membrane with flexibility, toughness, self-healing ability, and thermal stability. At a 1:1 ratio of GE and SA, the polymer electrolytes exhibited the highest tensile strength, measuring 2.14 MPa, and the highest ionic conductivity, reaching 3.7 × 10^−2^ S cm^−1^. In terms of capacity, rate capability, and cycle stability, the assembled full cells utilizing a V_2_O_5_/CNT nanocomposite cathode and a Zn metal anode surpassed their aqueous electrolyte counterparts. The improved rate performances and cyclic stability could be ascribed to (1) better interfacial contact between electrodes and electrolyte due to the homogenous morphology; (2) high ionic conductivity as a result of the 3D cross-linked network structure and multiple functional groups; and (3) good elasticity and toughness of the polymer to bear the Zn-dendrite penetrations.

Zhuoyuan et al. synthesized zinc–alginate (ZA) GPE with compact and dense morphology through an ion exchange reaction approach [118]. The GPE demonstrated a high ionic conductivity (1.24 mS cm^−1^ at room temperature) and excellent ion migration behavior (t_Zn_^2+^ = 0.59). More importantly, the non-porous structure favored homogeneous ion flux, leading to smooth deposition of Zn and suppression of dendrites. The resulting cell with Ca_0.24_V_2_O_5_-based cathode had a long lifespan of over 600 cycles, with a capacity retention of 88.7% under 3 C conditions.

### 5.5. Gelatin-Based Electrolyte

Yu Liu’s group successfully fabricated a self-standing gelatin-based hydrogel electrolyte (GHE) by combining a mixture of 2 M ZnSO_4_ and 0.1 M MnSO_4_. The GHE was strengthened through a low-temperature cooling process, and then in situ coated onto the MnO_2_ cathode to assemble flexible solid-state zinc metal batteries [119]. By implementing the strengthened electrolyte, the formation of Zn dendrites was effectively suppressed, and the self-corrosion of Zn foil was significantly slowed down. GHE showed an order of magnitude smaller absolute value of the deposition current than the aqueous electrolyte in the chronoamperometry test, indicating reduced dendrite on the anode surface, as seen in Figure 3A. The galvanostatic tests on the Zn//Zn symmetric cell, as presented in Figure 3B, indicated a much longer dendrite-free cycling time of GHE (over 800 h at 0.2 mA cm^−2^ and 1200 h at 0.1 mA cm^−2^) compared with aqueous electrolyte, although GHE had a smaller voltage hysteresis. The SEM images of the Zn foil after cycling (Figure 3C) further proved the GHE confinement on zinc anode self-corrosion and dendrites formation.

In addition, Yu Liu’s group employed a simple soaking strategy to dehydrate the gelatin-based solid-state electrolyte (GSE). This was followed by treatment in concentrated inorganic salt solutions, which resulted in the formation of strong hydrophobic interactions between gelatin molecular chains and enhanced bundling of chains within the gel network. The GSE exhibited a significantly enhanced mechanical strength, with an increase of 1–2 orders of magnitude, reaching 2.78 MPa. This mechanical strength was found to be the highest among all the single-component and self-standing solid-state Zn^2+^-conducting electrolyte systems [120]. Under extreme working conditions, the solid-state Zn/MnO_2_ full cell demonstrated excellent performance. It delivered a high reversible specific capacity of 285 mAh g^−1^, showed superior cycling stability with 90% capacity retention after 500 cycles, and exhibited outstanding safety.

Besides the increased mechanical strength, the gelation-based composite electrolyte also had good electrochemical properties. Chunyi Zhi’s group fabricated a novel gelatin and PAM-based hierarchical polymer electrolyte (HPE) for wearable solid-state ZIBs [98]. Through the grafting of PAM onto gelatin chains and filling in a PAN network, the HPE demonstrated exceptional performance. It exhibited high zinc ion conductivity of 1.76 × 10^−2^ S cm^−1^, along with high areal energy density and power density (6.18 mWh cm^−2^ and 148.2 mW cm^−2^, respectively). Furthermore, the HPE demonstrated exceptional flexibility and exhibited excellent cyclic stability, retaining 97% of its capacity after 1000 cycles at 2772 mA g^−1^. These remarkable characteristics make it an extremely promising material.

A gelatin/ZnSO_4_/glutaraldehyde electrolyte was designed to be employed in a novel in-plane interdigitated configuration of ZIB with the V_5_O_12_·6H_2_O cathode, which could avoid the utilization of separator, binder, and metallic current collector and greatly enhance the energy density [121]. A high capacity of 556 mAh/g was achieved at 0.1 A/g, with no noticeable difference under bending conditions, demonstrating its flexibility.

### 5.6. Nafion Membrane and Filter Paper

#### 5.6.1. MnO_2_ Cathode

Yuan et al. [122] tried to solve the zinc dendrite problem by proposing a porous lignin@Nafion composite membrane as a separator. In the study, effective SEI layer formation on the zinc anode was investigated and was mainly composed of zinc hydroxide sulfate (ZnSO_4_ [Zn(OH)_2_]_3_ *x* H_2_O, ZHS) during the plating–deposition process in mild electrolytes, as shown in Figure 4A. Figure 4B showed the initial growth of a water-rich SEI phase with high resistance on the Zn surface, where its signal (at approximately 8.5° in the XRD patterns) formed after the 10th cycle and became stronger after the 100th cycle. Comparatively, Nafion and lignin@Nafion membranes enabled the formation of a better planar SEI, i.e., water-deficient ZHS with much smaller resistance, and the deposition of Zn (0 0 2) parallel to the surface, which was different from the possible exposure of deposited Zn (1 0 0) and loose layer of ZHS protrudes using physical separator systems such as filter paper. The -SO^3−^ of Nafion interacted with -Zn^2+^ ion, which coordinated the remodeling of Zn^2+^ according to the DFT research. Furthermore, the addition of lignin enriched water channels and rendered subsequent high ionic conductivity. The Nafion separator showed an impressive life cycle of approximately 345 h in the plating–deposition test, surpassing that of commercial physical separators by several factors. Moreover, by incorporating 10 wt.% lignin, the life cycle could be further extended to around 410 h.

#### 5.6.2. Vanadium-Based Cathode

Wang’s group [123] examined the issue of zinc dendrite formation by analyzing various separators, such as filter membranes, filter paper, and glass fiber separators. Their findings indicated that the use of low-cost filter membrane separators, which have a uniform distribution of pore structure and flexibility, significantly enhanced the uniform deposition of zinc. Consequently, the cell incorporating the filter membrane demonstrated excellent cycling performance at different current densities. Moreover, the filter membrane exhibited the longest lifespan of approximately 2600 h and a minimal voltage hysteresis of 47 mV in the zinc symmetrical cell test conducted at a current density of 1 mA cm^−2^. The Zn//NaV_3_O_8_·1.5H_2_O cell demonstrated exceptional cycling stability, lasting more than 5000 cycles while retaining 83.8% of its initial capacity under the current density of 5 A g^−1^. The remarkable ability of the filter membrane to inhibit dendrite formation can be attributed to its uniform pore distribution and reliable mechanical properties, rather than its composition, as revealed by the mechanism exploration.

Ghosh et al. [124] fabricated a Nafion ionomer membrane with integrated Zn^2+^, which effectively inhibited the growth of irregular zinc deposits on the metal anode in water-based Zn/V_2_O_5_ batteries, thus promoting the cycling stability of the cell compared with the batteries using porous separators. The Zn^2+^ coordinated with the SO^−3^ in Nafion could lead to the transfer of a high number of Zn^2+^ cations, hence promoting ionic conductivity. As a result, the Nafion ionomer membrane contributed to improved plating and stripping behavior of zinc due to its higher ionic conductivity, low activation energy, low R_ct_ value, and enhanced interfacial compatibility with the electrodes. The corresponding Zn//V_2_O_5_ cell delivered a high capacity of 510 mAh g^−1^ at 0.25 A g^−1^. 

#### 5.6.3. Organic Cathode

Extensive research has been conducted on organic materials as alternative cathodes for electrochemical energy storage devices. These materials offer several advantages such as low toxicity and sustainability compared to conventional inorganic cathode materials. Additionally, they have exhibited remarkable performance in both monovalent and divalent cation storage, showing their superior capabilities. Quinones, which are widely present in nature, have emerged as sustainable and environmentally friendly electrode materials. Their versatile properties make them suitable for use in electrochemical energy storage devices. Zhao et al. [125] tested the different performances of calix[4]quinone (C4Q) in aqueous ZIBs with two different separators: filter paper and Nafion membrane. The presence of a Nafion membrane in the ZIB played a crucial role in enhancing its cycling performance [126]. This membrane exhibited ion-selective conducting ability, effectively inhibiting cross-over reactions between the electrode materials, Zn^2+^ ions, and C4Q^2*x*−^ ions. As a result, the battery demonstrated improved stability, preventing the generation of undesirable by-products and preserving the morphology of the Zn anode. The ZIB achieved a high specific capacity of 335 mAh g^−1^ at 20 mA g^−1^ and an impressive lifespan of 1000 cycles with a capacity retention of 87%.

### 5.7. Other Electrolytes

Wang et al. [127] presented a novel solid-state electrolyte called water@ZnMOF-808 (WZM), which consisted of a crystalline single-ion Zn^2+^ component, as shown in Figure 5A. This electrolyte was derived from a post-synthetic modified metal–organic framework (MOF) material, specifically ZnMOF-808. The ZnMOF-808 served as a host with fixed anionic microporous properties, while the Zn^2+^ ions within the electrolyte remained mobile. The SSE exhibited excellent properties, including a high ionic conductivity of 2.1 × 10^−4^ S cm^−1^ at 30 °C, a low activation energy of 0.12 eV, and a high Zn^2+^ transfer number of 0.93, and demonstrated good mechanical and electrochemical stability. The effective suppression of dendrite on zinc anode (Figure 5B,C) can be explained by the different Zn deposition behaviors/mechanisms of the MOF-based solid electrolyte that guide the Zn ion transportation and confine the Zn(H_2_O)_6_^2+^ ions through nano-wetted Zn/electrolyte interface, which differed from those of the ZnSO_4_ aqueous electrolyte-based counterpart (Figure 5D). The XRD results (Figure 5E) evidenced this confinement effect on Zn(H_2_O)_6_^2+^ ions. In 250 cycles, the VS_2_/Zn battery demonstrated a reversible capacity of 125 mAh g^−1^ at 0.2 A g^−1^.

To address the issue of insufficient ion conductivity, a covalent organic framework (COF)-based hydrogel electrolyte was prepared with sulfonic acid group modification to weaken the binding between Zn^2+^ and cations and promote ion dissolution, leading to ionic conductivity as high as 27.2 S cm^−1^ and a transference number of 0.89, as well as good mechanical strength [128]. The corresponding ZIB showed cyclic stability and an excellent rate capacity of 90 mAh g^−1^ at 10 C.

Sivaraman et al. [129] added organically modified clay to build a PEO/ZnSO_4_/nanoclay electrolyte. The resulting material exhibited the highest ionic conductivity when the weight ratio of PEO/ZnSO_4_/nanoclay was 10/5/0.75, reaching a value of 5.54 × 10^−4^ S cm^−1^. The utilization of a Zn–PANI gel electrolyte in the battery enabled it to achieve a capacity of 43.9 Ah kg^−1^ and a specific energy density of 52.6 Wh kg^−1^. This highlights the impressive performance of the battery in terms of its energy storage capabilities.

The utilization of petroleum-based plastics in energy storage devices raises environmental concerns. A biodegradable GPE, composed solely of natural cellulose, was recently proposed for the ZIB technology [130]. The cellulose membrane had an ultrathin (10 μm) non-porous structure and high water absorption/restriction capabilities, resulting in not only a high ion transfer number of 0.46 but also light weight and low cost of the whole cell. Additionally, its excellent mechanical property effectively inhibited the wide growth of dendrites, i.e., 1800 h of zinc stripping/plating process at 1 mA cm^−2^. This study promotes the development of environment-friendly energy storage devices.

## 6. Solid-State Electrolyte-Based Hybrid-Ion Batteries

Compared with single-ion batteries, rechargeable hybrid-ion batteries have the advantages of high operating voltage and large energy density. By taking advantage of the superior active nature of zinc metal that makes it stably electrodeposited from an aqueous electrolyte and the low-cost feature, aqueous hybrid-ion batteries depending on the zinc anode have received widespread attention. Hybrid-ion batteries, similar to commercial rechargeable battery systems utilizing organic electrolytes, operate by facilitating the transfer of electrons through an external circuit, while also facilitating the movement of metal ions such as Li^+^, Na^+^, and Zn^2+^ through the electrolyte. The process of insertion/extraction of electrode materials is commonly accompanied by redox reactions. In most rechargeable hybrid-ion batteries, the reactions occurring at both the cathode and anode electrodes involve different ions. The electrolyte used in these batteries contains various electrochemically active ions. Unfortunately, this composition often leads to undesired chemical cross-over in the battery system.

Arumugam Manthiram’s group recently proposed a ‘mediator-ion’ strategy to use monovalent ions as shuttle ions to maintain a current circuit, therefore monovalent-ion-conductive solid electrolytes could be applied in aqueous hybrid electrolyte batteries [131]. To address the issue of undesired chemical cross-over, specific mediator ions are utilized to shuttle between the catholyte (the aqueous electrolyte at the cathode) and the anolyte (the aqueous electrolyte at the anode) through the solid-state electrolyte. This enables the maintenance of the redox couples involved in the battery’s reactions. This strategy not only mitigated the issue of chemical cross-over but also helped to alleviate the problem of metal dendrite formation on metal anodes. The electrode materials used in these batteries are not limited to the solid phase. Additionally, it is possible to employ different anolytes and catholytes in a single cell, thus significantly expanding the possibilities for optimizing the electrochemical properties of the batteries. A Sodium (Na) Super Ionic Conductor (NASICON)-type material with the composition Na_3.4_Sc_2_(PO_4_)_2.6_(SiO_4_)_0.4_ was synthesized and utilized as an ionic mediator for charge transfer between the cathode and the anode. This material facilitated the exchange of Na+ ions, enabling efficient electrochemical reactions within the battery system [132]. Along with another Li^+^ ion exchanging membrane (LATSP, Li_1+*x*+*y*_Al*_x_*Ti_2−*x*_P_3−*y*_Si*_y_*O_12_), the two solid-state electrolytes were tested in various batteries, including Zn(NaOH) || Na-SSE || Br_2_(NaBr), Zn(NaOH) || Na-SSE || K_3_Fe(CN)_6_(NaOH), Fe(NaOH) || Na-SSE || K_3_Fe(CN)_6_(NaOH), Zn(LiOH) || Li-SSE || Br_2_(LiBr), Zn(NaOH) || Na-SSE || air (H_3_PO_4_/NaH_2_PO_4_), and Zn(LiOH) || Li-SSE || air (H_3_PO_4_/LiH_2_PO_4_) batteries. A schematic diagram of aqueous batteries enabled via a mediator-ion solid electrolyte is presented in Figure 6A [133]. The use of solid electrolyte separators offers several advantages in battery systems. First, it helps prevent chemical cross-over and addresses the issue of metal dendrite formation. Additionally, it allows for the incorporation of different liquid electrolytes in the anode (anolyte) and cathode (catholyte) within a single battery, offering flexibility and optimization of electrochemical properties. Despite this, there are still areas that require improvement. The low ionic conductivity of the solid electrolyte material and the mechanical instability of large-area solid electrolytes are challenges that the solid electrolyte community must address through collaborative efforts. Future advancements in these areas will contribute to the overall development and enhancement of solid electrolyte technology. Chen et al. [134] designed a Zn/KMnO_4_ double-electrolyte cell thanks to the separation of the acidic electrolyte from the alkaline electrolyte by the super-ionic conductor (LATSP) separator. The EIS measurement (Figure 6B) demonstrated the strong ionic selective conductivity of protons (H^+^) and Li^+^ by LATSP, which allowed the transfer of Li^+^ but prevented that of H^+^. In the proposed Zn/KMnO_4_ cell, the open circuit voltage (OCV) was significantly increased, reaching a high value of 2.8 V. This voltage corresponds to the theoretical potential difference between a Zn/KMnO_4_ cathode in an acidic electrolyte and a Zn anode in an alkaline electrolyte. By achieving this OCV, the cell demonstrates the ability to harness the full potential energy of the respective cathode and anode materials. By expanding the electrochemically stable potential window to nearly 3 V, the aqueous electrolyte used in the Zn/KMnO_4_ battery has enabled a significantly higher theoretical energy density. This increase brings the energy density of the battery to a level comparable to that of LIBs based on organic electrolytes. A similar mechanism was applied in the Zn–Cu Daniell-type battery, which gave this traditional battery type new potentials. In a traditional Zn–Cu Daniell-type battery, Dong et al. [135] utilized a thin film of ceramic lithium super-ionic conductor (LATSP). The LATSP thin film successfully separated the Cu cathode and the Zn anode and only exchanged lithium ions (Li^+^) between two electrodes, which avoided the Cu^2+^ cross-over-induced self-discharge, thereby achieving rechargeable characteristic in the Zn–Cu Daniell-type battery. During the cycling of the assembled battery, it was observed that the battery retained its capacity without any noticeable attenuation for a total of 150 cycles. Additionally, the investigation of OCV revealed that the potential remained stable for more than 100 h without any capacity loss.

Rulin Dong’s group [137] used a PVDF/PMMA-LiClO_4_/PVDF sandwiched membrane to conduct only Li^+^ as the electrochemical messenger between two separate aqueous electrolytes (ZnSO_4_ and CuSO_4_) and fabricated the Zn–Cu Daniell-type battery with metallic Zn and Cu as the anode and cathode, respectively. The electrochemical window of the battery spanned from 0.5 to 1.5 V. At a current density of 1 mA cm^−2^, the battery exhibited a practical capacity of up to 330 mA h g^−1^ based on the mass of active cathode material (CuSO_4_). This practical capacity amounted to 98.5% of the theoretical capacity, showing the battery’s efficient utilization of its active materials.

Dominik P.J.Barz’s group [138] recently employed a low-price commercial Neosepta CIMS monovalent cation exchange membrane in combination with a sodium sulfate (Na_2_SO_4_) background electrolyte to construct a Zn–Cu Daniell-type battery. The Neosepta CIMS membrane exhibited selective permeability towards monovalent cations, allowing for the passage of Na^+^ while effectively blocking Cu^2+^. The battery underwent 100 charge and discharge cycles without experiencing significant degradation.

Liu et al. [136] recently enhanced the SSE by creating a dual network structure, as shown in Figure 6C. In this design, PAM served as the supportive framework, providing a solid base. Meanwhile, SA served as the second network, forming crosslinked chains in the presence of Zn^2+^ cations. Compared to the pure PAM hydrogel, the SA–PAM solid electrolyte demonstrated significant improvements in its tensile and compressive strength. Specifically, the tensile strength increased from 67.78 to 228.86 kPa, while the compressive strength increased from 0.74 to 6.54 MPa. The improved strength of the solid electrolyte can be attributed to the successful cross-linking of the dual networks. This cross-linking played a crucial role in effectively suppressing corrosion and dendrite formation at the zinc anode. The electrolyte displayed an impressive mixed ionic conductivity of 19.74 mS cm^−1^. This high conductivity played a crucial role in enabling the PB@CNT|Zn hybrid aqueous batteries to exhibit excellent rate performance and remarkable cycling stability. The capacity of the cell varied from 93.8 to 61.1 mAh g^−1^ from 0.5 C to 16 C and recovered back to 89 mAh g^−1^ at 0.5 C, much better than that of the liquid electrolyte-based cell. With a low-capacity loss of only 0.0027% per cycle over 10,000 cycles, the batteries demonstrated exceptional durability. This can be attributed to the electrolyte’s ability to restrict the presence of free water molecules, effectively inhibiting cathode dissolution by 79.5%.

Aqueous rechargeable zinc–iodine batteries (ARZIBs) have been intensively studied in the last few years and show superior sustainability by virtue of the highly compatible Zn–I_2_ pairs within aqueous electrolytes [139,140,141,142]. In addition, iodine possesses impressive characteristics as a cathode material, including a remarkable theoretical capacity of 211 mAh g^−1^ and a high redox potential of I_3_^−^/I^−^ (0.53 V vs. SHE). These properties make it one of the top-performing cathodes when combined with Zn metal. Notably, significant advancements in capacity have been achieved in both liquid and solid-state batteries. For instance, capacities of 109 mAh g^−1^ have been attained over 500 cycles [143], 174 mAh g^−1^ over 3000 cycles [144], and an impressive 210 mAh g^−1^ over 10,000 cycles [145]. Nevertheless, zinc–iodine batteries face challenges such as low utilization of solid iodine and self-discharge problems, which arise from the diffusive nature of soluble polyiodides. A recent study by Sonigara et al. [146] introduced a novel solid-state Zn–I_2_ battery design using an amphiphilic block copolymer (F77, PEO_53_–PPO_34_–PEO_53_). In this design, the catholyte and electrolyte gel were separately integrated into carbon cloth and cellulose, respectively. This unique configuration aimed to enhance iodine solubility, promote reactivity, and prevent diffusion of iodide through the microcrystalline structure. The incorporation of ion-selective hydrophilic/hydrophobic channels within the gel catholyte facilitated these improvements, offering a promising solution for the challenges associated with zinc–iodine batteries. The solid-state Zn–I_2_ battery had a discharge capacity of 210 mAh g^−1^ at 1 C and 94.3% capacity retention after 500 cycles.

## 7. Advanced Functional Solid-State ZIBs

In addition to traditional batteries that pursue high energy, power density, and extended life cycles, more advanced functional requirements have been proposed by the academic community and application market. Solid-state batteries have the possibility of providing new functions because of their novel electrochemical properties. Among them, the polymer electrolyte can enhance the performance of the hydrogel electrolyte, e.g., strong tolerance to temperature/deformation and self-repair through the introduction of the functional groups of the polymer chain.

Zhao et al. [147] suggested the use of triblock PEO-PPO-PEO (Pluronic) copolymers, composed of poly(ethylene oxide) (PEO) and poly(propylene oxide) (PPO), as a thermo-reversible hydrogel to fabricate a smart flexible ZIB with cooling recovery capability. Unlike conventional flexible batteries, when this battery suffers extreme deformations, in situ fractures can be repaired (healing efficiency of up to 98 %) through a cooling process (fast recovery within 5 min), which restores the electrochemical performance and leads to increased lifespan and durability in unfavorable mechanical environments.

Aqueous batteries with conventional gel electrolytes that have high water content are often plagued by low-temperature performances due to the frozen electrolyte problem that occurs at subzero temperatures. In the coolant market, low molecular vicinal alcohols like glycerol and ethylene glycol (EG) are widely employed as non-toxic inhibitors to prevent water freezing [148,149]. Zhi et al. [150] developed a novel dual-crosslinked hydrogel by synthesizing EG-based waterborne anionic polyurethane acrylates (EG-waPUA) using a step-growth polymerization method [151]. Subsequently, they utilized a free radical polymerization technique to copolymerize the EG-waPUA precursor and AM monomers, resulting in the formation of an EG-waPUA/PAM-based hydrogel with dual crosslinking. The strong binding energies between water molecules and the hydroxyl groups of EG-waPUA, as well as the carbonyl groups of PAM chains, resulted in enhanced interactions. These interactions effectively trapped water molecules within the polymer networks and hindered the formation of crystalline structures. As a result, the hydrogel exhibited exceptional anti-freezing properties. By utilizing the developed electrolyte, an anti-freezing Zn–MnO_2_ battery was constructed, demonstrating impressive performance at low temperatures. The battery achieved a high specific capacity of 226 mAh g^−1^ at a rate of 0.2 A g^−1^ in −20 °C conditions, which corresponded to approximately 82% of the capacity achieved at room temperature. Furthermore, the battery exhibited excellent capacity retention, retaining approximately 72.54% of its initial capacity value after 600 cycles at a rate of 2.4 A g^−1^. In contrast, ZIB with a traditional PAM-gel electrolyte quickly failed at −20 °C due to electrolyte freezing.

Chen et al. [152] made modifications to conventional PVA gel electrolytes by crosslinking PVA and glycerol in the presence of borax, resulting in a modified gel electrolyte denoted as PVA-B-G. The inclusion of glycerol in the gel electrolyte created strong interactions with PVA chains, effectively preventing the formation of ice crystals, thereby reducing the freezing point [153,154]. This gel electrolyte had a freezing point below −60 °C, enabling it to function effectively even in extremely cold environments. Even at −35 °C, the gel electrolyte maintained a high ionic conductivity of 10.1 mS cm^−1^, as shown in Figure 7A, showing its excellent ability to facilitate the flow of ions. Additionally, the gel electrolyte demonstrated remarkable mechanical properties, further enhancing its overall performance. Using this anti-freezing gel electrolyte as a foundation, a flexible quasi-solid-state aqueous Zn–MnO_2_ battery was constructed. The battery demonstrated an impressive energy density of 46.8 mWh cm^−3^ (1.33 mWh cm^−2^) at a power density of 96 mW cm^−3^ (2.7 mW cm^−2^) at room temperature. It also exhibited excellent cycling durability, with approximately 90% capacity retention over 2000 cycles, thanks to the exceptional compatibility of PVA-B-G with the Zn anode. Furthermore, the battery displayed remarkable tolerance to various extreme conditions, even at −35 °C, as depicted in Figure 7B.

Chen et al. [155] generated a PAM/graphene oxide/EG gel electrolyte with the goal of enhancing ionic conductivity, mechanical properties, and anti-freezing capability of PAM gel electrolytes. This was achieved by leveraging the synergistic effect of EG and GO. The gel electrolyte displayed excellent electrochemical durability and exceptional tolerance to extreme working conditions in the assembled flexible Zn–MnO_2_ battery. This was attributed to its high ionic conductivity of 14.9 mS cm^−1^ at −20 °C, enabling optimal performance.

Wang et al. [156] developed a sodium polyacrylate hydrogel (PANa) electrolyte with ultra absorption of high ion concentrations (6 M OH^−^). High ion concentrations could effectively decrease the formation and content of hydrogen bonds, thus endowing a great anti-freezing property. Besides, with the decrease in water content due to the high ionic concentration, the density of molecular chains per unit section was much higher, providing enhanced tensile strength and strain. The hydrogel could be stretched to 1400% in both cold (−20 °C) and hot (50 °C) conditions. The assembled flexible NiCo//Zn battery could operate from −20 to 50 °C, delivering a high energy density of 172 Wh kg^−1^ at −20 °C and 210 Wh kg^−1^ at 50 °C, as well as favorable cycling performance.

**Figure 7 polymers-15-04047-f007:**
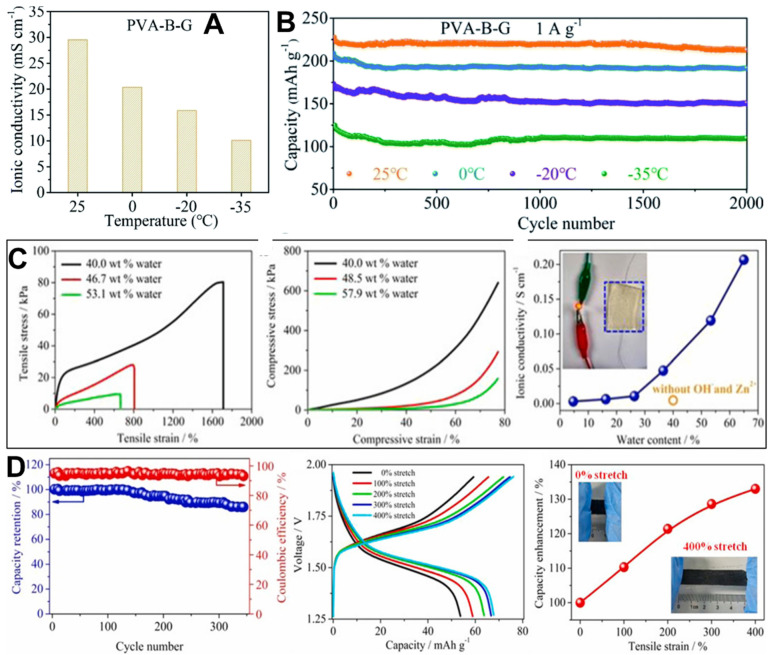
(**A**). The ionic conductivities of PVA-B-G with respect to temperature. (**B**). Cyclic performance of the PVA-B-G enabled batteries at 1 A g^−1^ at various temperatures. Reproduced from Ref. [152] with permission from the Royal Society of Chemistry. (**C**). Stress-strain curves of the PANa hydrogel with various water content under (**left**) stretching and (**middle**) compression. (**right**) Ionic conductivities of the PANa hydrogel with various water and ion content. The photograph shows an inset of the PANa hydrogel film, which effectively acts as a proficient ionic conductor, effectively linking the LED circuit. (**D**). (**left**) Cycling performance at 56.5 C. (**middle**) Charging–discharging curves at 23 A g^−1^ from 0% to 400% strain. (**right**) Capacity enhancement with respect to the tensile strain. Insets depict photographs of the battery when completely discharged and when stretched to a strain of 400%. Reprinted from Ref. [157], with permission from Elsevier.

Table 2 presents a summary of other reported developments of solid-state zinc rechargeable batteries that have been applied in a wide temperature range.

In addition, equipment with high mechanical properties (stretchable, compressible) have a wide range of potential applications. At present, there are many studies focused on expanding the mechanical properties of solid electrolytes through various means. Liu et al. [157] introduced a PANa electrolyte-based NiCo//Zn battery that exhibited remarkable properties. This battery had the ability to stretch by 400% and compress by 50% while still maintaining functionality. Furthermore, it endured 500 cycles of stretching and 1500 cycles of compression, retaining 87% and 97% of its initial capacity, respectively. Figure 7C illustrates the high ionic conductivity of the PANa hydrogel infused with a concentrated aqueous electrolyte (65% water content), recorded at 0.2 S cm^−1^. The battery achieved notable cyclic stability, as evidenced by an 86% capacity retention after 340 cycles and a 93% Coulombic efficiency at an extreme 56.5 C-rate, as depicted in Figure 7D. During the stretching process, the reduction in uncontacted areas between the electrolyte and electrode resulted in greater involvement of effective electrode materials in the redox process. This led to a significant enhancement in capacity, increasing it by 1.3 times.

## 8. Summary and Outlook

Significant efforts have been dedicated to studying active electrode materials and aqueous electrolytes in the field of aqueous zinc-ion batteries (AZIBs), leading to notable advancements in understanding their electrochemical behaviors. In contrast, the development of solid-state electrolytes for zinc-ion batteries (ZIBs) is still in its early stages. In this review, we discussed recent advancements in solid-state electrolytes for ZIBs, covering aspects such as membrane separators and inorganic and organic solid-state electrolytes.

Solid-state electrolytes offer compelling advantages over aqueous electrolytes, including reduced water content, which leads to improved confinement of cathode materials and stable electrochemical performance. They also effectively address issues associated with traditional aqueous electrolytes, such as leakage and electrolyte evaporation. However, challenges such as relatively low ionic conductivity and mechanical stability persist.

To overcome these challenges and facilitate the commercialization of ZIBs, future research should focus on enhancing the conductivity and mechanical properties of solid electrolytes. Achieving high ionic conductivity comparable to conventional liquid electrolytes is a primary challenge. It is crucial to develop solid electrolytes with improved conductivity while maintaining stability and compatibility with electrode materials to enhance overall performance and energy density. Ensuring the mechanical stability of solid electrolytes is equally critical, as mechanical stresses during charge and discharge cycles can degrade electrode–electrolyte interfaces, leading to reduced performance and cycle life.

One promising approach is the development of composite solid electrolytes by incorporating nanostructured materials such as carbon-based additives, ceramic salt fillers, inorganic perovskites (e.g., BaTiO_3_, PbTiO_3_, and LiNbO_3_), high dielectric constant fillers, and ionic liquids. These additions can enhance both ionic conductivity and mechanical strength. Advanced characterization techniques, such as in-situ spectroscopy and imaging, are valuable tools for gaining insights into the electrochemical processes occurring in solid-state electrolytes, enabling the design of tailored materials for improved performance.

Cost-effective synthesis and production of high-quality solid electrolytes at scale are crucial for commercialization. Developing efficient and sustainable manufacturing processes, along with optimizing material synthesis routes, will reduce costs and improve the economic viability of ZIB technology. Exploring advanced manufacturing techniques, such as scalable thin-film deposition or additive manufacturing, allows for precise control over the composition, thickness, cross-linking, and morphology of solid electrolytes, enhancing scalability and cost-effectiveness. Regulatory considerations and safety assessments are essential to ensuring the long-term stability and safety of solid-state electrolytes under various operating conditions, addressing concerns like dendrite formation and thermal runaway.

The future direction of ZIB research involves further advancements in solid-state electrolytes to enhance performance and stability. Addressing challenges related to ionic conductivity, mechanical stability, scalable manufacturing, and safety considerations is crucial for the successful commercialization of ZIB technology. Collaboration between academia, industry, and regulatory bodies, coupled with concerted research efforts, can propel ZIBs toward becoming a competitive and sustainable energy storage solution, potentially complementing or replacing lithium-ion batteries across diverse applications.

## Figures and Tables

**Figure 1 polymers-15-04047-f001:**
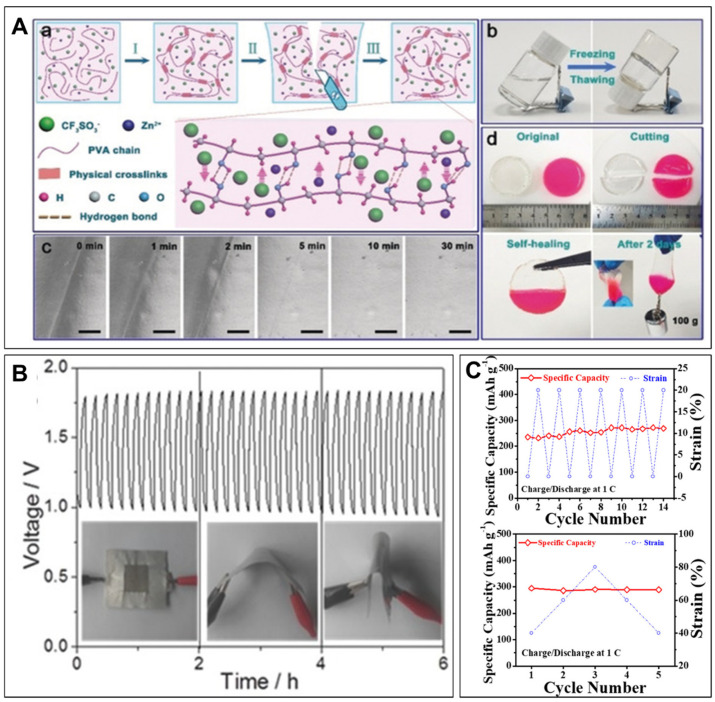
(**A**). (**a**) The fabrication process of the self-healing PVA/Zn(CF_3_SO_3_)_2_ hydrogel electrolyte and its self-healing illustration; Optical images of (**b**) hydrogel electrolyte before and after freezing/thawing, (**c**) self-healing process and (**d**) the self-healing behavior of the hydrogel electrolyte [52] © 2019 Wiley-VCH Verlag GmbH & Co. KGaA, Weinheim. (**B**). Galvanostatic discharge–charge cycling curves of the all-solid-state rechargeable Zn–air battery at 2 mA cm^−2^ with NCNF-1000 serving as the catalyst. A bending strain was applied every 2 h during the cycling process [58]. © 2016 WILEY-VCH Verlag GmbH & Co. KGaA, Weinheim. (**C**). Compressibility of the rechargeable Zn–MnO_2_ battery with PAM gel electrolyte: the specific capacities of the battery under cyclic compressional strain (**top**) from 0 to 20% and (**bottom**) from 40 to 80% and then return to 40%. Reprinted with permission from [59]. Copyright 2018 American Chemical Society.

**Figure 3 polymers-15-04047-f003:**
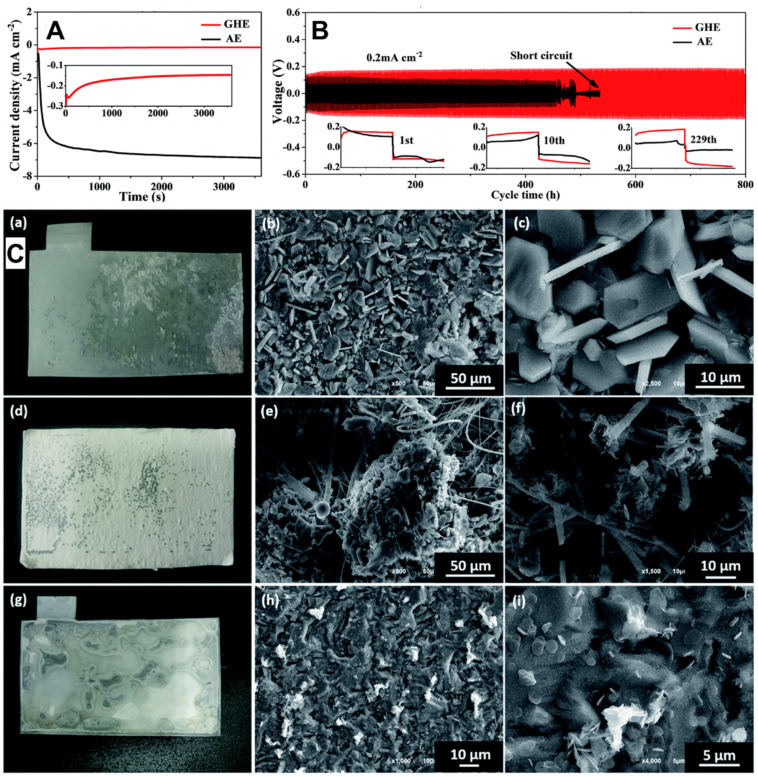
(**A**). Comparison of the chronoamperometric curves in the aqueous electrolyte (AE) and gelatin hydrogel electrolyte (GHE) within a three-electrode system. (**B**). Galvanostatic voltages of Zn//Zn symmetric cells assembled with AE or GHE at 0.2 mA cm^−2^ and 0.2 mAh cm^−2^. (**C**). Pictures of battery components after cycling: (**a**) Zn electrode in the AE, (**b**) absorbent glass mat in the AE, and (**g**) Zn electrode in the GHE; SEM images of the counterparts: (**b**,**c**) surface of (**a**), (**e**,**f**) surface and cross-section of (**d**), and the (**h**,**i**) surface of (**g**). Reproduced from Ref. [119] with permission from the Royal Society of Chemistry.

**Figure 4 polymers-15-04047-f004:**
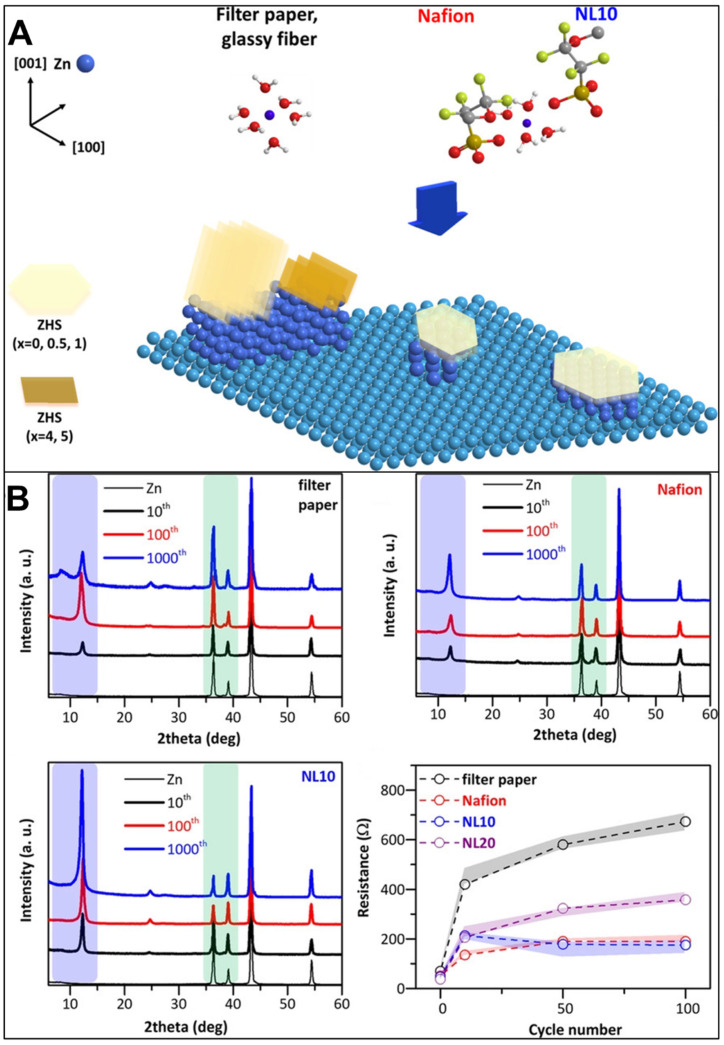
(**A**). Proposed scheme summarizing the effects of membranes on the surface of Zn metal. The interaction between –SO_3_^−^ groups from Nafion and Zn^2+^ ions potentially leads to a modification in the coordination of Zn^2+^ ions from their original state in the electrolyte (Zn^2+^(H_2_O)_6_). Consequently, this alteration in Zn^2+^ coordination results in the formation of contrasting growth modes for ZHS and deposited Zn. (**B**). Ex situ grazing-incidence XRD patterns for Zn metal at various plating–deposition cycles using (**top left**) filter paper, (**top right**) Nafion, and (**bottom left**) NL10 at 0.2 mA cm^−2^. (**bottom right**) variation in interfacial resistance during the initial 100 cycles of symmetric cells with different separators/membranes. The bands corresponding to the curves served as a reference to determine the measured data ranges [122] © 2019 Wiley-VCH Verlag GmbH & Co. KGaA, Weinheim.

**Figure 5 polymers-15-04047-f005:**
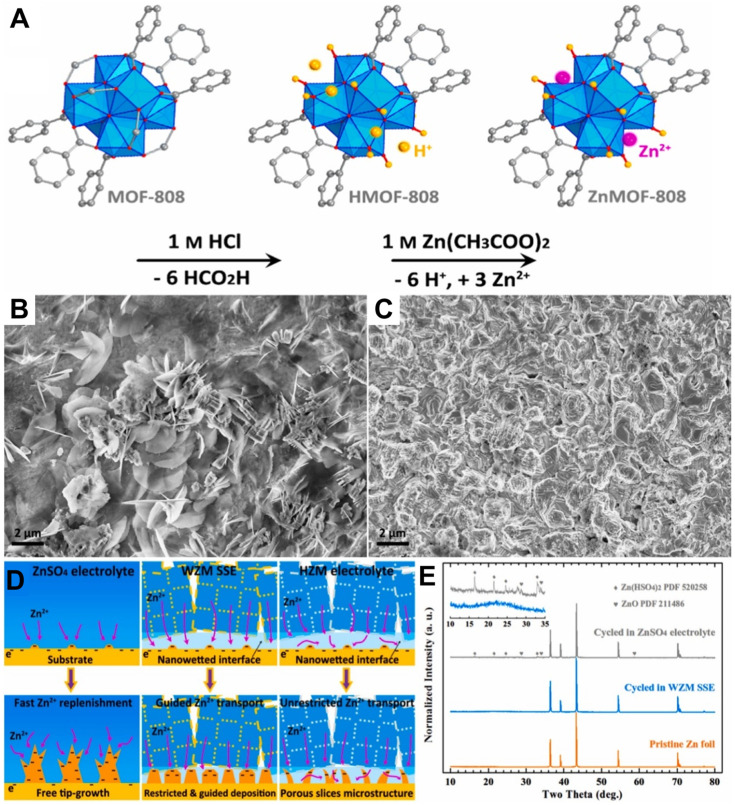
(**A**). Scheme of the post-synthetic modification chemistry. SEM images of the Zn foils after plating/stripping cycles of the (**B**) Zn/ZnSO_4_/Zn cell and the (**C**) Zn/WZM/Zn cell. (**D**). Schematics of the Zn^2+^ deposition processes with (**left column**) ZnSO_4_ aqueous electrolyte, (**middle column**) WZM SSE, and (**right column**) hybrid ZnSO_4_@MOF-808 electrolyte. (**E**). XRD patterns of the Zn foils before (orange) and after plating–deposition processes in WZM SSE (blue) and ZnSO_4_ electrolyte (grey), respectively. Reprinted from Ref. [127], with permission from Elsevier.

**Figure 6 polymers-15-04047-f006:**
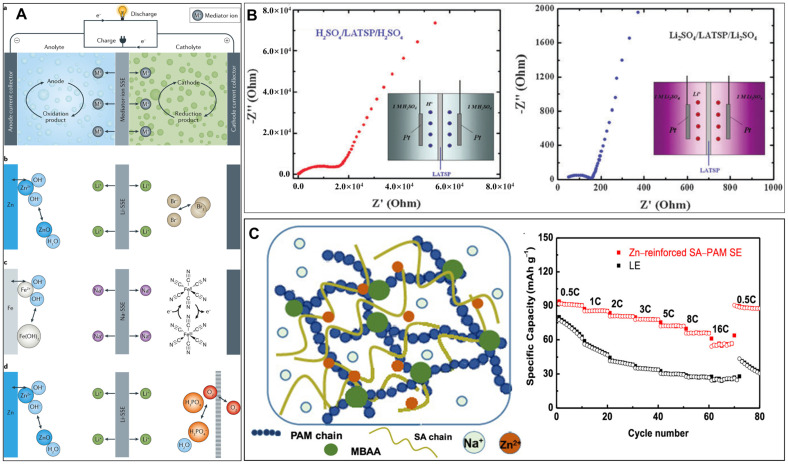
(**A**). (**a**) Schematic of aqueous batteries with mediator-ion solid electrolytes; (**b**) Zn-anode-based, (**c**) Fe-anode-based and (**d**) Zn−O_2_ aqueous battery systems with mediator-ion solid electrolytes. Reproduced from Ref. [133] with permission from Springer Nature. (**B**). Nyquist plot obtained from EIS measurements of the (**left**) H_2_SO_4_–LATSP–H_2_SO_4_ and (**right**) Li_2_SO_4_–LATSP–Li_2_SO_4_ systems. Reproduced from Ref. [134] with permission from the Royal Society of Chemistry. (**C**). (**left**) Schematic of the evolution of the hydrogel structure, where navy lines and khaki lines represent PAM chains and alginate chains, respectively, green dots represent covalent cross-links (MBAA), while the rosy dots represent ionic cross-links (Zn^2+^); (**right**) rate performances of Na–Zn hybrid batteries with Zn-reinforced SA–PAM SE and traditional LE. Reprinted with permission from [136]. Copyright 2020, American Chemical Society.

**Table 1 polymers-15-04047-t001:** Summary of other proposed solid-state alkaline zinc rechargeable batteries at room temperature.

Electrolyte	Ionic Conductivity (mS cm^−1^)	Energy/Power Density	Cyclic Performance	Reference
KOH-doped PVA	15	581 Wh kg^−1^	120 cycles at 50 mA g^−1^	[79]
Quaternary ammonia (QA)-functionalized nanocellulose	23	492 mAh g^−1^	200 cycles at 250 mA g^−1^	[80]
Laminated nanocellulose/GO membrane with QA	33.3	-	30 cycles at 1 mA cm^−2^	[81]
KOH-doped PVA/PAA nanofiber membrane	11.2	-	250 cycles at 20 mA cm^−2^	[82]
QA modified PVA	23.1	223 Wh kg^−1^	120 cycles at 1 mA cm^−2^	[83]
KI-PVA-PAA-GO	155	742 mAh g^−1^	20 cycles at 2 mA cm^−2^	[84]
PVA-GG-GA-PCL	123	11.87 Wh kg^−1^	100 cycles at 2 mA cm^−2^	[85]
KOH-doped PAM	215.6	720 mAh g^−1^	140 cycles at 5 mA cm^−2^	[86]

**Table 2 polymers-15-04047-t002:** Summary of solid-state electrolyte applied in a wide temperature range.

Electrolyte	Specific Capacity	Cyclic Performance	Reference
PAM/glycerol/acetonitrile/ZnSO_4_	262 mAh g^−1^ @ 500 mA g^−1^/60 °C 138 mAh g^−1^ @ 500 mA g^−1^/−20 °C	500 cycles @ 60 °C 500 cycles @ −20 °C	[158]
Sodium polyacrylate/6 M electrolyte	125 mAh g^−1^ @ 16 C/50 °C 120 mAh g^−1^ @ 19 C/−20 °C	900 cycles @ 50 °C 10000 cycles @ −20 °C	[156]
PAM–cellulose nanofiber hydrogel	300 mAh g^−1^ @ 500 mA g^−1^/50 °C 267 mAh g^−1^ @ 500 mA g^−1^/−18 °C	100 cycles @ −18 °C	[159]
6 M ZnCl_2_/PVA	171 µAh cm^−2^ @ 1.0 mA cm^−2^/80 °C	-	[160]
Acetamide/zinc perchlorate hexahydrate/PAM	250 mAh g^−1^ @ 50 °C	30 cycles @ 50 °C	[161]
PAM/ZnSO_4_/Glycerol/xanthan gum	125 mAh g^−1^ @ 200 mA g^−1^/0 °C	-	[108]
PAM cross-linked with DMSO and cellulose nanofibers	118 mAh g^−1^ @ 200 mA g^−1^/−20 °C	350 cycles @ −20 °C	[162]
PAM/SA/PMMA nanoparticles	37 mW cm^−2^ @ −20 °C (Zinc-air)	-	[163]
PAM/modified polysaccharides carboxymethyl chitosan/Zn(ClO_4_)_2_ salts	70 mAh g^−1^ @ 5 A g^−1^/−30 °C	2500 cycles @ −30 °C	[164]
Sorbitol-modified cellulose hydrogel/ 16 M ZnCl_2_	64.4 mAh g^−1^ @ 1 A g^−1^/−40 °C	1000 cycles @ −40 °C	[165]

## Data Availability

No new data were created.

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
