# Peer review of "A Minireview of the Solid-State Electrolytes for Zinc Batteries"

_polymers, 2023, doi:10.3390/polym15204047_

Round 1

Reviewer 1 Report

This review article fairly discusses the recent developments of various material concepts for zinc ion batteries. The authors highlighted the pros and cons of different electrolytes and their compatibility with particular electrode materials. The technical details in terms of battery performance outlined in this review seem to be useful for material selection for future research of ZIBs. I would recommend this article to be published with minor revisions. The following suggestions are given to further improve the review. 

1. Section 4 should be re-organized. Some sub-section doesn’t seem to match with the title of section 4. For instance, though the main focus appears to be about electrolyte, the authors have included the discussion of cathode and other materials. Also, this section is too long which makes it difficult to follow. I would recommend it condense or re-organize with appropriate subtitles.

2. I would also recommend the authors to elaborate the discussion regarding the future direction of ZIB’s research and challenges associated with these materials in commercializing.

Reviewer 2 Report

The review is devoted to the fabrication and study of solid electrolytes for various types of zinc-ion batteries. It may be published after taking into account the following comments:

1) A separate small section is needed to describe methods for preparing polymer and gel-polymer membranes

2) Are electrode materials composite? If they contain a polymer electrolyte, then it’s also worth devoting a couple of paragraphs to this

3) A separate small section should be devoted to the high and low temperature properties of various types of electrolytes. It is highly advisable to add a table with examples, especially for low temperatures

4) It is necessary to indicate at what temperature the conductivity values were obtained in Table 1

5) Figure 5 is stretched horizontally

6) Some images use fragments with low resolution

7) There is a lot of unnecessary information in Figure 6 (for example, part C), and the concept of a hybrid battery is shown very vaguely. It is proposed to make the graphical explanation of the text more clear

8) The abstract states that inorganic electrolytes will be mentioned, but the article only gives examples of lithium and sodium super-ionic conductors. Are there inorganic solid electrolytes for zinc?

Reviewer 3 Report

I have carefully assessed the manuscript titled "A Minireview on Solid-State Electrolytes for Zinc Batteries." I find this review to be an intriguing and comprehensive exploration of the subject matter. Nevertheless, I have certain reservations regarding its suitability for publication.

Firstly, while the content is engaging, it is imperative to acknowledge that the literature already boasts a plethora of similar reviews on this topic. Additionally, the references cited by the authors predominantly pertain to sources published prior to 2022. In light of the rapidly evolving landscape of research in this field, this aspect raises concerns about the paper's currency and relevance.

In light of the aforementioned considerations, I must regrettably recommend rejecting the submitted manuscript in its current form. However, I am open to reconsidering this decision if the authors choose to revise and update the review with more recent references, thereby ensuring its alignment with the latest developments in this field. This course of action would significantly enhance the manuscript's suitability for resubmission and potential publication in the journal.

Round 2

Reviewer 2 Report

The authors have made all necessary changes to the text of the article. The work can be published

Reviewer 3 Report

It can be now accepted for publication.